# Fall of Community-Acquired Pneumonia in Children following COVID-19 Non-Pharmaceutical Interventions: A Time Series Analysis

**DOI:** 10.3390/pathogens10111375

**Published:** 2021-10-24

**Authors:** Alexis Rybak, David Dawei Yang, Cécile Schrimpf, Romain Guedj, Corinne Levy, Robert Cohen, Vincent Gajdos, Julie Tort, David Skurnik, Naïm Ouldali, François Angoulvant

**Affiliations:** 1Assistance Publique-Hôpitaux de Paris, Pediatric Emergency Department, Robert Debré University Hospital, Université de Paris, 75019 Paris, France; alexis.rybak@aphp.fr; 2INSERM UMR 1123, ECEVE, Université de Paris, 75019 Paris, France; naim.ouldali@aphp.fr; 3ACTIV, Association Clinique et Thérapeutique Infantile du Val-de-Marne, 94000 Créteil, France; corinne.levy@activ-france.fr (C.L.); robert.cohen@activ-france.fr (R.C.); 4Assistance Publique-Hôpitaux de Paris, Pediatric Emergency Department, Necker-Enfants Malades University Hospital, Université de Paris, 75015 Paris, France; daviddawei.yang@aphp.fr (D.D.Y.); cecile.schrimpf@aphp.fr (C.S.); 5Assistance Publique-Hôpitaux de Paris, Pediatric Emergency Department, Armand Trousseau University Hospital, Sorbonne Université, 75012 Paris, France; romain.guedj@aphp.fr; 6Obstetrical, Perinatal and Pediatric Epidemiology Research Team (EPOPé), Centre de Recherche Épidémiologie et Statistique Sorbonne Paris Cité (CRESS), Inserm UMR 1153, Université de Paris, 75004 Paris, France; 7Centre Hospitalier Intercommunal, Research Centre, Université Paris Est, IMRB-GRC GEMINI, 94000 Créteil, France; 8Assistance Publique-Hôpitaux de Paris, Pediatric Department, Antoine Béclère University Hospital, Université de Paris Saclay, 92140 Clamart, France; vincent.gajdos@aphp.fr; 9Center for Research in Epidemiology and Population Health, INSERM UMR 1018, 94800 Villejuif, France; 10Assistance Publique-Hôpitaux de Paris, Patient Quality Medical Organisation Departement-Health Crisis Management, 75004 Paris, France; julie.tort@aphp.fr; 11Assistance Publique-Hôpitaux de Paris, Department of Clinical Microbiology, Necker-Enfants Malades University Hospital, Université de Paris, 75015 Paris, France; david.skurnik@aphp.fr; 12INSERM U1151-Equipe 1, Institut Necker-Enfants Malades, Université de Paris, 75015 Paris, France; 13Division of Infectious Diseases, Harvard Medical School, Boston, MA 02115, USA; 14Assistance Publique-Hôpitaux de Paris, Department of General Pediatrics, Pediatric Infectious Disease and Internal Medicine, Robert Debré University Hospital, Université de Paris, 75019 Paris, France; 15INSERM, Centre de Recherche des Cordeliers, UMRS 1138, Sorbonne Université, Université de Paris, 75006 Paris, France; 16Inria, HeKA, Inria, 75006 Paris, France

**Keywords:** community-acquired pneumonia, children, COVID-19, non-pharmaceutical interventions, mitigation measures, time series analysis

## Abstract

Non-pharmaceutical interventions (NPIs) were implemented to reduce the spread of coronavirus disease 2019 (COVID-19). A first national lockdown was decided in France on the 17 March 2020. These measures had an impact on other viral and non-viral infectious diseases. We aimed to assess this impact on community-acquired pneumonia (CAP) in children. We performed a quasi-experimental interrupted time series analysis. We used data from a French prospective surveillance system of six pediatric emergency departments (PEDs). All visits from 1 January 2017 to 31 December 2020 were included. Pre-intervention period was before 17 March 2020 and post-intervention period was after 18 March 2020. We estimated the impact on the weekly number of visits for CAP and CAP admission using quasi-Poisson regression modeling. A total of 981,782 PEDs visits were analyzed; among them, 8318 visits were associated with CAP, and 1774 of these were followed by a hospital admission. A major decrease was observed for CAP visits (−79.7% 95% CI [−84.3; −73.8]; *p* < 0.0001), and CAP admission (−71.3% 95 CI [−78.8; −61.1]; *p* < 0.0001). We observed a dramatic decrease of CAP in children following NPIs implementation. Further studies are required to assess the long-term impact of these measures.

## 1. Introduction

Before the vaccination programme at the end of 2020, the fight against the COVID-19 epidemic mainly relied on preventive measures. These non-pharmaceutical interventions (NPIs) combined social distancing, especially in stores and transport, wearing of face masks, and reinforcing hand washing with hydro-alcoholic solution and soap. These measures were reinforced in France by population confinement between 17 March and 11 May 2020 and from 30 October to 15 December 2020 [1,2,3]. Schools were closed during the first confinement but remained open during the second with a health protocol including mandatory wear of masks for children from the age of six years.

As a secondary benefit, a dramatic decrease of viral pediatric respiratory infection, such as bronchiolitis, has been observed worldwide [4,5,6]. This fall was correlated with a sharp decrease of pediatric emergency department (PED) visits and hospital admissions [2,7,8]. The most plausible explanation for this observed reduction was the interruption of person-to-person viral respiratory transmission. Interestingly, this decrease has also affected other diseases such as asthma exacerbation [9,10], and acute otitis media [11] which highlight the role of a viral trigger.

Community-acquired pneumonia (CAP) in children is a major cause of hospitalization and of death among children younger than five years [12]. CAP can be caused by various pathogens: viruses, bacteria, and fungi. Among these, *Streptococcus pneumoniae* is the most frequently isolated bacteria, especially in severe and fatal pediatric forms [12]. Recently, a global decrease of invasive pneumococcal disease that has been reported during the pandemic was attributed to a reduced person-to-person transmission of *S. pneumoniae* [13].

We hypothesized that NPIs would have a large impact on CAP in children, especially for the most severe forms. To address this question, we conducted a quasi-experimental interrupted time series analysis based on multicenter prospective French surveillance data for PED visits and related hospital admissions. The study objective was to describe the association between NPI during the SARS-CoV-2 pandemic and PED visits associated with CAP.

## 2. Results

A total of 981,782 PEDs visits in the six participating centers from 1 January 2017 to 31 December 2020 were included. Among them, 91,087 were followed by a hospital admission. Overall, 8318 PED visits were diagnosed as CAP, and 1774 of these were followed by a hospital admission. Data collected before March 2020 were used to generate a model fitting the observed values of the PEDs visit, admission and CAPs, allowing us to project the number that could have been expected without NPIs.

As presented in Figure 1 and Table 1, a dramatic decrease was observed between expected and observed values since the implementation of NPIs in March 2020 for CAP visits (−79.7% 95% CI [−84.3; −73.8]; *p* < 0.0001), and CAP admission (−71.3% 95 CI [−78.8; −61.1]; *p* < 0.0001). The usual seasonal peak of CAP observed every year between 2017 and 2019 in autumn and winter has not been observed in 2020. The low frequency of CAP diagnoses was observed even when the NPIs rules were relaxed between June and October 2020.

We also found a sharp decrease of PEDs visits and related hospital admission between expected and observed values since NPIs introduction: (−42.3% 95 CI [−47.2; −37.1]; *p* < 0.0001), and (−28.9% 95 CI [−33.8; −23.7]; *p* < 0.0001) respectively (Figure 2). The observed decrease of PEDs visits and related admission seem to be of a lesser magnitude than the CAPs visits and admission decrease.

## 3. Discussion

In this time series analysis of 981,782 PEDs visits, the number of PED visits related to CAP, and CAP admission decreased by −80% and −71%, respectively, in 2020 after the establishment of NPIs. The decrease of overall PEDs visits and related hospital admission was also massive, reaching −42% and −29%, respectively, although of lesser magnitude. Our findings provided additional argument, suggesting the decrease in person-to-person transmission of respiratory pathogens, both viruses and bacteria, with NPIs [5,6,7,13].

Looking more closely at these data, we found that the second French lockdown, from 31 October to 15 December 2020, also had a major impact on CAP, PEDs visits and related hospital admission. During this period, daycare centers and schools remained open, and only adult and children older than 6 years old were mandated to wear a face mask [1,14]. This underlines the importance of adults and older children in the transmission dynamics of these respiratory pathogens.

Defining the pathogen responsible of a CAP is in most cases a challenge. The pathophysiology of bacterial CAP in children is complex. While viral infection is not an essential step to bacterial CAP, it seems highly frequent and facilitates pneumococcal infections, particularly for less invasive serotypes [15,16,17]. Pneumococcus is suspected to be responsible in an important proportion of CAP, particularly in hospitalized patients [18]. The global decrease of IPD incidence, with an estimated decrease of 77% eight weeks after lockdown in France, suggests that CAP due to pneumococcus decreased [13]. It is important to specify that despite the major change in health seeking during 2020, vaccination against pneumococcus remained high in children [19]. Viral epidemiology has been highly affected by NPIs. The French National Agency (Santé Publique France) reported a diminished respiratory syncytial outbreak in 2020 [20]. The peak of bronchiolitis occurred after a three-month delay and was about 50% as high as previous seasons [21]. The same study reported a stable positivity rate of rhinovirus in patients with bronchiolitis [21]. Furthermore, the influenza epidemic has been slightly shortened in 2020, with NPI being implemented at the very end of the epidemic [22]. However, the change in CAP associated with an RSV, influenza or rhinovirus infection have not been reported. The other respiratory viruses, such as parainfluenza and adenovirus, are not monitored in France, despite the wide use of multiplex PCR. In conclusion, these epidemiological changes may have impacted the viral infections but also the viral–bacterium co-infections. Our results also suggest that COVID-19 mostly spares children. Our data did not allow us to differentiate the decrease due to a possible lower transmission of pneumococcus from the decrease in respiratory viruses. The massive fall observed in CAP hospitalizations suggests that the entire spectrum of CAP has fallen and not just the mild viral forms.

Other countries reported a massive decrease in pediatric CAP after NPIs implementation [23,24]. Among these countries, some experienced an important total duration of school closure (57 weeks in Brazil), whereas other decided to limit this measure (12 weeks in France and 6 weeks in Switzerland) [25]. These experiences suggest that non-restrictive NPIs, i.e., without schools and day-care center closure, may have an important positive impact on pediatric CAP.

Our study has several limitations. We cannot exclude a change in clinical management such as avoidance of pharynx examination because of COVID-19 fear, which could have influenced diagnosis coding; apart from admission, we did not collect data regarding severity and so we cannot exclude that reduction in presentations was associated with children presenting later in their illness. While the dramatic decrease in PEDs could be partially due to transportation limitations, a fear of going to the hospital, and an increase of telemedicine, the difference between the evolution of PED visits and CAP, and the significant decrease in CAP admission, do not favor this hypothesis. Our study relies on discharge codes, i.e., data recorded in daily practice which mean that some misclassifications occurred at the individual level. However, the method of coding was expected to be constant over time, thus at a population level it is unlikely that the changes observed in the trends observed could be explained by misclassification. Aggregated data did not include clinical information other than the hospital admission or discharge, thus we could not analyze clinical symptoms at presentation. The analysis is limited to the period from January 2017 to December 2020. Our study period did not include the annual RSV outbreak, which was delayed to February–March 2021, but the bronchiolitis outbreak was massively reduced compared to previous seasons [20,21]. For influenza, no outbreak has been observed during the 2020/2021 season [22]. Moreover, our model takes into account seasonality [26].

## 4. Conclusions

We observed a marked reduction both in PED visits for CAP, and in CAP followed by a hospital admission in children, following NPI implementation in France. This reduction was persistent after partial relaxation of the mitigation strategies. Further studies are required to determine the long-term impact of these public health interventions on airborne transmitted pathogens.

## 5. Materials and Methods

### 5.1. Source of Data

The Regional Centre of Observation and Action on Emergencies e-CERVEAU, Agence Régionale de Santé, is an official network of emergency departments dedicated to public health that automatically transmit a summary of anonymized data from all their visits to the regional database. The database has been approved by the French data protection authority (CNIL). These data include discharge diagnosis coded by the physicians in charge of the patient at the end of the visit according to the International Statistical Classification of Diseases and Related Health Problems (10th revision) and hospital admission or discharge. This study covers six PEDs from academic hospitals being part of Assistance Publique—Hôpitaux de Paris, located in and around Paris. These centers annually gather 250,000 visits and transmitted their data daily from 1 January 2017 to 31 December 2020. We used the e-CERVEAU database for this research. Patient informed consent is not required according to current dispositions. For this study, visits with one of the following codes were considered as pneumonia and extracted: A370; A371; A378; A379; J13; J14; J150; J151; J152; J153; J154; J155; J156; J157; J158; J159; J16; J160; J168; J17; J170; J172; J173; J178; J180; J181; J182; J188; J189; J200; J201; J202; J690; J849; J851; J860; J869; and J90. Visits were grouped by calendar weeks for each year. 

### 5.2. Interventions in France

Between 17 March 2020 and 11 May 2020, the first strict lockdown with daycare and school closure was decided in France. Daycare and schools reopened from 11 May 2020, initially for children younger than 11 years old, and then progressively all children. From 1 September 2020, the wearing of face masks was compulsory during school attendance for children older than 11 years. This measure was extended to adults working in daycare centers and schools from 18 September 2020 and to children older than six years old at school, since 2 November 2020. Between 31 October 2020 and 15 December 2020, a second lockdown, without daycare center or school closure, was decided. This lockdown was relayed by a night curfew. The detailed NPIs implemented in France are detailed by the European Centre for Disease Prevention and Control [14].

### 5.3. Outcome Measure

The main outcome was the weekly number of PED visits for pneumonia over time. The secondary outcomes were the weekly number of PED admissions for pneumonia over time, the weekly number of PED visits over time, and the weekly number of PED visits followed by an admission over time. Visits and hospital admissions for urinary tract infections (UTI) served as a control outcome, as NPIs are not expected to impact on these pathologies [2].

### 5.4. Statistical Analysis

Outcomes were analyzed by quasi-Poisson regression, accounting for seasonality, secular trend before and after lockdown, and overdispersion of data [26,27,28,29]. Seasonality was taken into account by including harmonic terms (sines and cosines) with 12-month and 6-month periods to adjust for the seasonal pattern [29]. The time unit chosen was one week to provide optimal precision to the model [26]. We hypothesized that the intervention would have an impact on the next time unit considering the incubation time of most viral diseases. Thus, intervention assessment involved a dummy variable in the model estimating the immediate post-intervention change [26,28]. Pre-intervention period was from 1 January 2017 to 17 March 2020 and post-intervention period was from 18 March 2020 to 31 December 2020. Intervention impact was estimated by comparing estimates in the post-intervention period to expected estimates if the lockdown did not occur, based on the quasi-Poisson regression model. We decided to analyze crude numbers rather than number per 1.000 PED visits because NPI had a major impact not limited to infectious diseases [2,7,8].Validity of the quasi-Poisson regression model was assessed by visual inspection of the correlograms (autocorrelation and partial autocorrelation functions) and residuals analysis. All statistical tests were two-sided, and we considered a result as “significant” when the *p*-value was < 0.05. All statistical analysis involved using Stata (StataCorp. 2019. Stata Statistical Software: Release 16. College Station, TX: StataCorp LLC).

## Figures and Tables

**Figure 1 pathogens-10-01375-f001:**
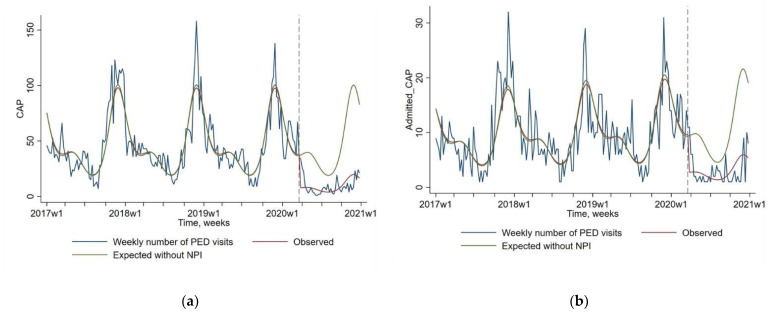
Impact of non-pharmaceutical interventions (NPIs) on weekly number of pediatric emergency department (PED) visits for community-acquired pneumonia or CAP (Figure (**a**), N = 8318), and for CAP followed by a hospital admission (Figure (**b**), N = 1774). The blue line shows the observed data. The red line shows the model estimates based on observed data (quasi-Poisson regression modeling). The green line shows the expected values without NPIs (quasi-Poisson regression modeling). The start of the NPIs is indicated by a vertical dashed line.

**Figure 2 pathogens-10-01375-f002:**
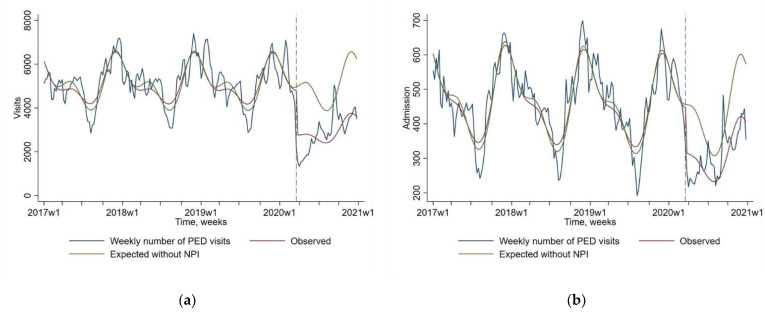
Impact of non-pharmaceutical interventions (NPIs) on weekly number of pediatric emergency department (PED) visits (Figure (**a**), N = 981,782) and for PED visits followed by a hospital admission (Figure (**b**), N = 91,087). The blue line shows the observed data. The red line shows the model estimates based on observed data (quasi-Poisson regression modeling). The green line shows the expected values without NPIs (quasi-Poisson regression modeling). The start of the NPIs is indicated by a vertical dashed line.

**Table 1 pathogens-10-01375-t001:** Changes in pediatric emergency departments visits and related admission in 2020 before and after non-pharmaceutical interventions.

	Percentage of Change Following NPIs [95% CI] ^1^	*p*-Value
Visits	−42.3 [−47.2; −37.1]	<0.0001
Admissions	−28.9 [−33.8; −23.7]	<0.0001
CAP	−79.7 [−84.3; −73.8]	<0.0001
Admitted pneumonia	−71.3 [−78.8; −61.1]	<0.0001

Abbreviations: CAP, community-acquired pneumonia; NPI, non-pharmaceutical interventions. ^1^ The change estimated by comparing estimates in overall post-intervention period to expected estimates if lockdown did not occur using quasi-Poisson regression modeling.

## Data Availability

Data are not available.

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
