# Peer review of "Fall of Community-Acquired Pneumonia in Children following COVID-19 Non-Pharmaceutical Interventions: A Time Series Analysis"

_pathogens, 2021, doi:10.3390/pathogens10111375_

Round 1

Reviewer 1 Report

This is a good piece of work that provides a dramatic illustration of the effect of NPIs on non-COVID respiratory infections. Although not completely novel (similar effects have been observed worldwide in other contexts), it really adds to this body of evidence nicely, by employing robust analytic methods and showing such an enormous effect. It will be a nice piece to cite in future.

The opportunistic interrupted time series design is appropriate, especially in light of the large effect size seen. 

The one thing I would like to see is a more detailed consideration of relative contribution of specific viral (and non-viral) microbial pathogens in the discussion. Although detailing specific microbial aetiology was beyond the scope of this study, most developed countries have systems in place for surveillance of respiratory pathogens (including at minimum pneumococcus, influenza and RSV - but also picornaviruses - including rhinoviruses, parainfluenza, adenovirus etc). These pathogens are detected on multiplex diagnostic platforms commonly used in routine care. One would think a country the size of France has aggregate data of this sort publicly available. So I would like to see much more contextual detail here that can help us explain the observations - what has happened in France to reported rates of invasive pneumococcal disease, influenza notifications and RSV (mentioned briefly but more detail needed) during the period of observation? This may shed some light on which pathogens are playing the greatest relative role in the phenomenon seen. For example some parts of the world have seen an almost complete interruption of influenza transmission, but with some residual RSV and rhinovirus transmission continuing. What was the situation in France?

Reviewer 2 Report

The authors utilized discharge diagnosis codes from a database of emergency departments to conduct an interrupted time series analysis of weekly PED visits and hospital admission for CAP before and after non-pharmaceutical interventions such as school closures in March 2020 due to COVID-19. They found that ED visits and hospitalizations for CAP decreased post-intervention even when NPIs were relaxed between June and October 2020 and overall PED visits and hospital admission for all-causes decreased as well. The authors highlight an interesting trend the COVID-19 pandemic and subsequent mitigation strategies have had on pediatric respiratory viral infections. One concern with the study is the limited time period of analysis, especially given the subsequent resurgence of respiratory viral infections particularly RSV that has been described in Australia and is currently occurring in the US. It would be helpful to increase the post-intervention period to capture more recent data which may better reflect the trends of pediatric CAP and healthcare visits during COVID-19. Also, the authors should mention that a limitation of the paper is reliance on discharge codes given the variability in the definition of CAP amongst healthcare providers, it is possible some patients were misclassified. Having pathogen-specific data or clinical symptoms at presentation would add greatly to the significance of this paper. In the current form, the paper provides a very limited snapshot of trends in pediatric healthcare utilization during the COVID-19 pandemic which has already been described.

Round 2

Reviewer 2 Report

I appreciate the authors taking time to revise portions of the manuscript. No additional comments.